# Effects of Environmental Exposure to Cadmium and Lead on the Risks of Diabetes and Kidney Dysfunction

**DOI:** 10.3390/ijerph19042259

**Published:** 2022-02-16

**Authors:** Supabhorn Yimthiang, Phisit Pouyfung, Tanaporn Khamphaya, Saruda Kuraeiad, Paleeratana Wongrith, David A. Vesey, Glenda C. Gobe, Soisungwan Satarug

**Affiliations:** 1Occupational Health and Safety, School of Public Health, Walailak University, Nakhon Si Thammarat 80160, Thailand; ksupapor@mail.wu.ac.th (S.Y.); phisit.po@mail.wu.ac.th (P.P.); tanaporn.kh@mail.wu.ac.th (T.K.); 2Medical Technology, School of Allied Health Sciences, Walailak University, Nakhon Si Thammarat 80160, Thailand; saruda.ku@wu.ac.th; 3Community Public Health, School of Public Health, Walailak University, Nakhon Si Thammarat 80160, Thailand; paleeratana.wo@wu.ac.th; 4Kidney Disease Research Collaborative, Centre for Health Services Research, The University of Queensland Translational Research Institute, Brisbane 4102, Australia; david.vesey@health.qld.gov.au (D.A.V.); g.gobe@uq.edu.au (G.C.G.); 5Department of Nephrology, Princess Alexandra Hospital, Brisbane 4075, Australia; 6School of Biomedical Sciences, The University of Queensland, Brisbane 4072, Australia; 7NHMRC Centre of Research Excellence for CKD, QLD, UQ Health Sciences, Royal Brisbane, and Women’s Hospital, Brisbane 4029, Australia

**Keywords:** cadmium, lead, albuminuria, diabetes, fasting plasma glucose, glomerular filtration rate

## Abstract

Environmental exposure to cadmium (Cd) or lead (Pb) is independently associated with increased risks of type 2 diabetes, and chronic kidney disease. The aim of this study was to examine the effects of concurrent exposure to these toxic metals on the risks of diabetes and kidney functional impairment. The Cd and Pb exposure levels among study subjects were low to moderate, evident from the means for blood concentrations of Cd and Pb ([Cd]_b_ and [Pb]_b_) of 0.59 µg/L and 4.67 µg/dL, respectively. Of 176 study subjects (mean age 60), 71 (40.3%) had abnormally high fasting plasma glucose levels. Based on their [Cd]_b_ and [Pb]_b_, 53, 71, and 52 subjects were assigned to Cd and Pb exposure profiles 1, 2, and 3, respectively. The diagnosis of diabetes was increased by 4.2-fold in those with an exposure profile 3 (*p* = 0.002), and by 2.9-fold in those with the estimated glomerular filtration (eGFR) ≤ 60 mL/min/1.73 m^2^ (*p* = 0.029). The prevalence odds ratio (POR) for albuminuria was increased by 5-fold in those with plasma glucose levels above kidney threshold of 180 mg/dL (*p* = 0.014), and by 3.1-fold in those with low eGFR) (*p* = 0.050). Collectively, these findings suggest that the Cd and Pb exposure profiles equally impact kidney function and diabetes risk.

## 1. Introduction

Chronic environmental exposure to cadmium (Cd) has been associated with increased risks of prediabetes and diabetes in the general populations in the U.S. [1,2,3,4], Korea [5], and China [6]. Two Norwegian population studies reported exposure to Cd and lead (Pb) as potential risk factors for diabetes [7,8]. Low environmental exposure to Cd has been identified as a risk factor for chronic kidney disease (CKD) in cross-sectional studies in Spain [9], Korea [10], and the U.S. [11,12,13,14,15]. The diagnosis of CKD is based on albumin-to-creatinine ratio (ACR) above 30 mg/g creatinine (albuminuria) and/or a decrease in the glomerular filtration rate (GFR) to 60 mL/min/1.73 m^2^ (low eGFR) that persists for at least three months [16,17,18,19]. In the Dallas lead project, an elevation of blood Pb ([Pb]_b_) was associated with a marked reduction in eGFR among residents of a lead smelter community [20]. In prospective cohort studies in Sweden, low environmental Pb exposure was linked to a GFR decrease, CKD onset, and end-stage kidney disease (ESKD) [21,22].

Simultaneous exposure to Cd and Pb is commonly encountered by the general populations of various nations [23,24], including the U.S. [25,26,27,28], Canada [29], Taiwan [30], and Korea [31]. In the U.S. National Health and Nutrition Examination Surveys (NHANES) 2007–2012, half of the participants, aged ≥ 6 years had [Cd]_b_ and [Pb]_b_ or urinary excretion rates of Cd and Pb above population median levels [27]. It was noted in the Cadmibel study that people with diabetes were more susceptible than those with no diabetes to Cd toxicity in kidneys [32]. A similar observation was subsequently made in women from Sweden [33,34], the Torres Strait, Australia [35], Korea [36], and the U.S. [13].

The aim of this study was to investigate the effects of concurrent exposure to Cd and Pb in 176 Thai subjects of which 88 were diagnosed diabetes, and 88 were apparently healthy non-diabetics. [Cd]_b_ and [Pb]_b_ levels were used as exposure estimates, while estimated GFR (eGFR) and albumin-to-creatinine ratio (ACR) served as kidney functional measures. GFR is considered to indicate function of surviving nephrons [16,17,18]. In practice, the GFR is estimated from equations, including the Chronic Kidney Disease Epidemiology Collaboration (CKD-EPI) [17], and is reported as an estimated GFR (eGFR). ACR is primarily a function of glomerular membrane permeability, and kidney proximal tubular reabsorption [37].

## 2. Materials and Methods

### 2.1. Participants

This study used a purposive sampling method to recruit type 2 diabetic subjects together with age- and gender-matched non-diabetic controls of equal number from a local health center of Pakpoon Municipality, Nakhon Si Thammarat Province, Thailand. It was undertaken during June 2020 to May 2021. The inclusion criteria were residents of Pakpoon municipality, 40 years of age or older who were diagnosed with type 2 diabetes or were apparently healthy. The exclusion criteria were non-residents of Pakpoon municipality, pregnancy, breast-feeding, a hospital record or physician’s diagnosis of an advanced chronic disease. All subjects were provided with details of study objectives, study procedures, benefits, and potential risks, and they all provided their written informed consents prior to participation. The sociodemographic data, education attainment, occupation, health status, family history of diabetes, and smoking status were obtained by structured interview questionnaires. Diabetes was defined as plasma glucose [Glc]_p_ levels ≥ 126 mg/dL or a physician’s diagnosis. Hypertension was defined as systolic blood pressure ≥ 140 mmHg, or diastolic blood pressure ≥ 90 mmHg. After excluding subjects with missing data, 176 subjects (88 with a diabetes diagnosis and 88 apparently healthy, non-diabetic controls) were included in the present study.

### 2.2. Simultaneous Blood and Urine Sampling and Biochemical Analysis

Participants were instructed to fast overnight, and the collection of blood and urine samples was carried out at a local health center of Pakpoon Municipality in the morning of the following day. For glucose assay, blood samples were collected in tubes containing fluoride as an inhibitor of glycolysis. For an analysis of Cd and Pb, blood samples were collected in separate tubes containing ethylene diamine tetra-acetic acid (EDTA) as an anticoagulant. The blood and urine samples were kept on ice and transported within 1 h to the medical technology laboratory of Walailak University, where plasma samples were prepared for various biochemical analyses. The remainder plasma and whole blood samples were aliquoted as were urine samples, stored at −80 °C for later analysis. Fasting plasma glucose concentrations ([Glc]_p_) were measured to ascertain diabetes diagnosis and diabetes free stage of controls. The assay of plasma concentration of glucose was based on colorimetry. Assays of creatinine in urine and plasma ([cr]_u_, [cr]_p_]) were based on the Jaffe reaction. Urine concentration of albumin ([Alb]_u_) was determined using an immunoturbidimetric method.

### 2.3. Analysis of Blood Concentrations of Cd and Pb

Blood concentrations of Cd and Pb ([Cd]_b_, [Pb]_b_) were determined with the GBC System 5000 Graphite Furnace Atomic Absorption Spectrophotometer (GBC Scientific Equipment, Hampshire, IL, USA). Multielement standards were used to calibrate metal analysis (Merck KGaA, Darmstadt, Germany). Reference urine and whole blood metal control levels 1, 2, and 3 (Lyphocheck, Bio-Rad, Hercules, CA, USA) were used for quality control, analytical accuracy, and precision assurance. The analytical accuracy of metal detection was checked by an external quality assessment every 3 years. All test tubes, bottles, and pipettes used in metal analysis were acid-washed and rinsed thoroughly with deionized water. When [Cd]_b_ and [Pb]_b_ levels were less than their detection limits, the concentrations assigned were their detection limits divided by the square root of 2 [38]. Ninety-four subjects (53.4%) had [Pb]_b_ below the detection limit of 3 µg/dL, and 61 (34.6%) had [Cd]_b_ below the detection limit of 0.1 µg/L.

### 2.4. Toxic Metal Exposure Profiling 

To enable evaluation of the impacts of simultaneous exposure to Cd and Pb on the risks of diabetes and kidney dysfunction, exposure to these toxic metals was based on [Cd]_b_ and [Pb]_b_. Each subject was assigned to the exposure profile 1, 2, or 3 by comparing her/his [Cd]_b_ and [Pb]_b_ with the median for [Cd]_b_ of 0.3 µg/L and the median for [Pb]_b_ of 2.12 µg/dL. Given the sample size of 176, we used the median as a cutoff value to obtain subgroups with sufficient numbers of participants. Exposure profile 1 was defined as [Cd]_b_ *and* [Pb]_b_ ≤ medians. Exposure profile 2 was defined as [Cd]_b_ *or* [Pb]_b_ levels ≥ medians. Exposure profile 3 was defined as [Cd]_b_ *and* [Pb]_b_ > medians. There were 53, 71, and 52 subjects with exposure profiles 1, 2, and 3, respectively.

### 2.5. Estimated Glomerular Filtration Rates (eGFR) 

The glomerular filtration rate was estimated with CKD-EPI equations [17], which were validated by inulin clearance [17]. Male eGFR = 141 × [serum creatinine/0.9]^Y^ × 0.993^age^, where Y = −0.411 if serum creatinine ≤ 0.9 mg/dL, Y = −1.209 if serum creatinine > 0.9 mg/dL. Female eGFR = 144 × [serum creatinine/0.7]^Y^ × 0.993^age^, where Y = −0.329 if serum creatinine ≤ 0.7 mg/dL, Y = −1.209 if serum creatinine > 0.7 mg/dL. For dichotomous comparisons, CKD was defined as eGFR ≤ 60 mL/min/1.73 m^2^ and CKD stages 1, 2, 3a, 3b, 4, and 5 corresponded to eGFR of 90–119, 60–89, 45–59, 30−44, 15–29, and <15 mL/min/1.73 m^2^, respectively.

### 2.6. Statistical Analysis

Data were analyzed with IBM SPSS Statistics 21 (IBM Inc., New York, NY, USA). The Kruskal–Wallis test was used to assess differences in means among three exposure groups, and the Pearson chi-squared test was used to assess differences in percentages. The one-sample Kolmogorov–Smirnov test was used to identify departures of continuous variables from a normal distribution, and a base-10 logarithmic transformation was applied to variables that showed rightward skewing before they were subjected to parametric statistical analysis. The multivariable logistic regression analysis was used to determine the Prevalence Odds Ratio (POR) for dichotomized outcomes. Abnormal fasting plasma glucose was defined as [Glc]_p_ ≥ 110 mg/dL. Diabetes was diagnosed when fasting [Glc]_p_ were ≥126 mg/dL. The renal threshold for glucose was assumed to be [Glc]_p_ ≥ 180 mg/dL. Albuminuria was defined as a ACR ≥ 20 mg/g for men and ≥30 mg/g for women. The Pearson’s correlation analysis was used to assess the strength of correlation of [Glc]_p_ and other variables. The means for [Glc]_p_ adjusted for age, BMI, and interaction in groups of subjects were obtained by univariate/covariance analysis with Bonferroni correction in multiple comparisons. For all tests, *p*-values ≤ 0.05 were considered to indicate statistical significance.

## 3. Results

### 3.1. Characteristics of Participants

Among 176 participants, half were given type 2 diabetes diagnosis by their medical records, while 3 of 88 persons recruited as non-diabetic controls had fasting plasma glucose [Glc]_p_ ≥ 126 mg/dL, a diabetes diagnosis level (Table 1).

About half (52%) were hypertensive, and 9.7% were smokers. Over half (54%) were overweight and 10.8% were obese. The overall mean for age was 59.9 years, and mean body mass index (BMI) was 25.4 kg/m^2^. The overall mean blood Cd concentration ([Cd]_b_) was 0.59 µg/L, and the mean blood Pb concentration ([Pb]_b_) was 4.67 µg/dL with the median [Cd]_b_ as 0.3 µg/L and the median [Pb]_b_ as 2.12 µg/dL. The prevalence rate of eGFR) ≤ 60 mL/min/1.7 m^2^ (low eGFR) was 16.5%, while ACR of 30−299 and ≥300 mg/g creatinine were 17.4% and 4.2%, respectively. About half had fasting plasma glucose [Glc]_p_ within a normal range (<110 mg/dL), while 11.9%, 25.6%, and 13% of participants had [Glc]_p_ of 110−125, 126−179, and ≥180 mg/dL, respectively.

The % of participants with diabetes diagnosis was the highest (69.2%), middle (46.5%), and lowest (35.8%) in profiles 3, 2, and 1 (*p* = 0.028). Likewise, smoking was the highest (17.9%), middle (9.9%), and lowest (1.9%) in profiles 3, 2, and 1 (*p* = 0.028). The % of men and women across three exposure profiles was similar (*p* = 0.092) as was the % of hypertension (*p* = 0.067). The percentages of four BMI classes (thin, normal, overweight, and obesity) across three Cd and Pb exposure profiles did not differ.

The mean [Glc]_p_ was highest (150 mg/dL), middle (128 mg/dL), and lowest (118 mg/dL) in exposure profiles 3, 2, and 1 (*p* = 0.031). The means for age (*p* = 0.639), BMI (*p* = 0.426), and duration of diabetes (*p* = 0.910) did not differ nor did the means for [cr]_p_ (*p* = 0.133), eGFR (*p* = 0.269), ACR (*p* = 0.389), [Alb]_u_ (*p* = 0.564), and [cr]_u_ (*p* = 0.499). Likewise, the % of low eGFR, ACR of 30−299 mg/g creatinine, and ≥300 mg/g creatinine among groups did not differ (*p* = 0.169−0.786) nor did the % distribution of four [Glc]_p_ levels (*p* = 0.154−0.368).

### 3.2. Logistic Regression Analysis 

Results in Table 2 show that the POR for diabetes diagnosis did not associate with age, BMI, smoking status, hypertension, or gender (*p* = 0.115−0.457), but it increased by 3-fold in participants with profile 2 (POR 2.991, 95% CI: 1.317, 6.794, *p* = 0.009), and by 4-fold in those with profile 3 (POR 4.182, 95% CI: 1.725, 10.14, *p* = 0.002). The POR for diabetes diagnosis increased by 3-fold in those with eGFR ≤ 60 mL/min/1.73 m^2^ (POR 2.926, 95% CI: 1.115, 7.679, *p* = 0.029).

Results in Table 3 indicate that POR for [Glc]_p_ equal to or above the renal threshold for glucose of 180 mg/dL increased slightly with age (POR 1.066, 95% CI: 1.011,1.124, *p* = 0.018), while POR for [Glc]_p_ ≥ 110 mg/dL (*p* = 0.653) and for [Glc]_p_ ≥ 126 mg/dL (*p* = 0.073) did not associate with age. POR values for all three levels of abnormal [Glc]_p_ did not associate with BMI (*p* = 0.095) or smoking (*p* = −0.301). In the profile 2 group, an increment of POR for [Glc]_p_ ≥ 110 mg/dL did not reach statistical significance (*p* = 0.086). However, in this profile 2 group, the POR [Glc]_p_ ≥ 126 and ≥180 mg/dL rose, respectively, by 2.29- (95% POR 2.290, CI: 1.073, 4.885, *p* = 0.032) and 3.38-fold (95% CI: 1.136, 10.08, *p* = 0.029). In the profile 3 group, the POR for [Glc]_p_ ≥ 110, ≥ 126, and ≥180 mg/dL rose by 2.79- (95% CI: 1.228, 6.375, *p* = 0.014), 2.964- (95% CI:1.288, 6.822, *p* = 0.011), and 3.41-fold (95% CI: 1.061, 10.94, *p* = 0.039), respectively.

### 3.3. Correlation Analysis

Of eight variables tested, [Glc]_p_ correlated significantly with four variables, including age, [Pb]_b_, ACR, and exposure profiles (Table 4). An inverse correlation was seen between [Glc]_p_ and age (*r* = −0.193), while positive correlations were seen between [Glc]_p_ and [Pb]_b_ (*r* = 0.194), ACR (*r* = 0.288), and exposure profiles (*r* = 0.216).

### 3.4. Covariance Analysis of Fasting Plasma Glucose Variation

Figure 1 depicts results of an analysis of the variation in [Glc]_p_ across three exposure groups, and subgroups, stratified by exposure profiles and blood pressure status.

Because the prevalence of smoking among participants was low (9.7%) and there was a gender bias (of 18 smokers, one was woman), the mean for fasting plasma glucose derived for each Cd-Pb exposure profile was not adjusted for smoking. The means for fasting plasma glucose were adjusted for age and BMI. Because Cd and Pb both are cumulative toxicants, the body burden of these metals increases with age. Obesity (BMI > 30 kg/m^2^) is a known risk factor of diabetes and hypertension. Cd and Pb exposure profiles marginally accounted for the variation in [Glc]_p_ (*F* = 2.781, η^2^ = 0.033, *p* = 0.065) (Figure 1a). A subgroup analysis showed that [Glc]_p_ was elevated, especially in participants with profile 3 who also had hypertension (*p* = 0.033), compared to those of the same exposure profile whose blood pressure was within a normal range (Figure 1b).

Table 5 shows results of logistic regression analysis of albuminuria in relation to low eGFR and abnormal fasting [Glc]_p_ levels.

By logistic regression analysis (Table 5), the POR for albuminuria was increased by 5-fold in participants with [Glc]_p_ ≥ 180 mg/dL (POR 4.937, 95% CI: 1.373, 17.76, *p* = 0.014), compared with those with [Glc]_p_ < 110 mg/dL. It rose by 3-fold in those with low eGFR (POR 3.093, 95% CI: 1.002, 9.552, *p* = 0.050). POR for albuminuria fell by 70% in the normotensive relative to the hypertensive (POR 0.294, 95% CI: 0.114, 0.755, *p* = 0.011). It also fell by 84% in women compared with men (POR = 0.161, 95% CI: 0.046, 0.570, *p* = 0.005).

## 4. Discussion

In the present study, we examined the effects of simultaneous exposure to Cd and Pb on fasting plasma glucose levels, and clinical kidney functional measures, low eGFR, and albuminuria. In total, 65.4% of the cohort had [Cd]_b_ above the detection limit, while 46.5% had [Pb]_b_ above the detection limit. This suggests that Cd exposure was more widespread than exposure to Pb. There was a wider variation in Cd exposure levels than Pb among subjects. Those subjects in the profile 3 group, where the mean [Cd]_b_ was 1.05 µg/L, had 1.62- and 21-fold higher [Cd]_b_ than those with profiles 2 and 1, respectively. The [Cd]_b_ levels among those with profiles 2 and 3 were in those ranges found to be associated with increased risk of CKD in the representative U.S. population [13,14]. [Cd]_b_ levels ≥ 0.61 μg/L in adult participants in NHANES 2007–2012 were associated with 1.8- and 2.2-fold increases in risk of low eGFR and albuminuria, respectively [13]. [Cd]_b_ levels ≥ 0.53 μg/L in adult participants in NHANES 2011–2012 were associated with 2.21- and 2.04-fold increases in the risk of low eGFR and albuminuria, respectively [14].

Those subjects in the profile 3 group, where the mean [Pb]_b_ was 7.38 µg/dL, had 1.61- and 3.48-fold higher [Pb]_b_ than those with profiles 2 and 1, respectively. Compared to [Pb]_b_ of 0.5 μg/dL, the level that has not been found to be associated with an adverse effect in adults in any epidemiologic studies [39], the mean [Pb]_b_ recorded for this study group of 4.67 µg/dL was 10 times higher. Indeed, [Pb]_b_ levels ≥ 2.4 μg/dL in adult participants in NHANES 1999−2006 were associated with 1.56-fold increase in the risk of low eGFR [11]. [Pb]_b_ levels ≥ 3.3 μg/dL were associated with 1.49-fold increase in the incidence of CKD, while [Pb]_b_ of 7.6 μg/dL was associated with an increase in the risk of ESKD in prospective cohort studies of the Swedish population [21,22].

In the present study, BMI was not a significant predictor of diabetes diagnosis (Table 2), nor was it associated with abnormal fasting plasma glucose levels (Table 3). Similarly, obesity was not related to Cd and Pb exposure profile (Table 1). This observation was consistent with results of studies in the U.S. [40,41], Canada [42], Korea [5], and China [6] showing an inverse relationship between measures of obesity and Cd exposure estimates such as [Cd]_b_, [Cd]_u_, or E_Cd_. Among participants of NHANES 1999–2002, an inverse association between [Cd]_u_ and central obesity was observed [40], while an inverse association between [Cd]_b_ and BMI was seen in the NHANES 2003–2010 cycle [41]. Similarly, an inverse association between [Cd]_b_ and BMI was seen in non-smokers in the Canadian Health Survey 2007–2011 [42]. A negative association between Cd exposure and various obesity measures was seen in both men and women in a study of the indigenous population of Northern Québec, Canada, where obesity was highly prevalent [43]. An inverse association between [Cd]_b_ and BMI was noted in a group of Korean men, 40–70 years of age, showing a mean for [Cd]_b_ of 1.7 μg/L, and a mean for E_Cd_ of 2.13 μg/g creatinine [5], while an inverse association between E_Cd_ levels ≥ 2.95 µg/g creatinine and reduced risk of overweight was reported by a Chinese study [6]. Urinary Cd levels were inversely associated with height and BMI in a study of Flemish children [44]. An association between urinary Cd and a reduction in risk of obesity by 54% was seen in 6–19 year old children and adolescents, enrolled in NHANES 1999–2011 [45].

The prevalence of abnormally high fasting [Glc]_p_ in this study group was high, 50.6%. Three subjects recruited as non-diabetic controls had [Glc]_p_ levels higher than a diabetes diagnosis level. One of these three subjects had exposure profile 3, while the other two had exposure profile 1. For the whole group, one in four (25.6%) had [Glc]_p_ between 127 and 179 mg/dL, and 13.1% had [Glc]_p_ exceeding the kidney threshold for glucose of 180 mg/dL, and all these 23 subjects had Cd-Pb exposure profile 3. The severity of impaired glycemic control, and exposure outcome as albuminuria both were related to their exposure profiles and the presence of hypertension (Figure 1b). It should, however, be noted that hypertension can be a cause or a consequence of CKD since blood pressure rises when GFR falls. Exposure profile 3 was associated with a 3.4-fold increase in the POR for [Glc]_p_ ≥ 180 mg/dL (*p* = 0.039) (Table 3). The POR for albuminuria increased by 5-fold in those with [Glc]_p_ ≥ 180 mg/dL (Table 5). Our findings underscore the conclusion from the nationwide Thai diabetes cohort study that glycemic control was more effective than other measures, such as blood pressure control, to delay the progression of kidney disease in patients with type 2 diabetes [46]. Notably, however, to achieve the target glycemic control, avoidance of high dietary intake of Cd and Pb may also be necessary.

It is noteworthy that the worldwide rising incidence of type 2 diabetes mellitus has often been linked to increasing prevalence of obesity, but studies in various populations, described above, found an inverse association between Cd exposure estimates and body weight gain and other measures of obesity. Consequently, exposure to Cd, especially of dietary origin, appears to be a contributor to the global increase in prevalence of diabetes.

Herein, we provide, for the first time, data that link simultaneous exposure to Cd and Pb to increased risks of abnormally high fasting plasma glucose, and kidney dysfunction, evident from changes in GFR and albumin excretion rate. The reasons for the GFR reduction seen in people with diabetes have been not been investigated adequately although the potential involvement of environmental exposure to Cd and Pb has been suggested by cross-sectional studies and longitudinal cohort studies [11,12,13,14,15,21,22]. We hypothesize that the low eGFR seen in those with high and moderate exposure to Cd and Pb, was a consequence of Cd toxicity in proximal tubular cells, known to accumulate Cd [23,24,47]. Abundant evidence suggests that Cd inflicts tubular cell injury at a low intracellular concentration of Cd and intensifies as the concentration rises. Inflammation and fibrosis follow, nephrons are lost, and GFR falls [23,24,47,48,49,50]. After substantial proximal tubular injury has occurred, reabsorption of albumin decreases and its excretion exceeds the normal limit.

The majority of epidemiological studies have considered Cd or Pb exposure, independently. Hence, our knowledge of the combined nephrotoxicity of low environmental exposure to Cd plus Pb has been limited. In the joint effect analysis using NHANES 1999–2006 data [13], [Cd]_b_ levels ≥ 0.6 µg/L were found to be associated with increases in risk of low eGFR and albuminuria by 1.53- and 1.92-fold, respectively. The increases in risk for low eGFR and albuminuria rose, respectively, to 1.98, and 2.34 in those who had [Cd]_b_ ≥ 0.6 µg/L plus [Pb]_b_ ≥ 2.4 μg/dL.

Of note, a dose–response meta-analysis has shown that the risk of diabetes increases, when [Cd]_b_ levels rise to levels above 1 µg/L [51]. However, data from the present study, and various NHANES cycles, discussed above, have linked [Cd]_b_ levels of 0.5−0.6 µg/L to increases in the risk of GFR reduction and albuminuria. Thus, Cd-induced nephropathy may contribute, in part, to an elevation of plasma glucose levels among the subjects in the present study, which accentuates the central role of kidneys in the maintenance of plasma glucose levels [52,53]. These notions are in line with experimental studies in rats, where daily subcutaneous doses of Cd at 0.6 mg/kg for 12 weeks caused proteinuria [54], and hyperglycemia [55]. Half of the subjects in the present study were diagnosed with type 2 diabetes (Table 2), and those with Cd and Pb exposure profiles 2 and 3 faced a 3–4.2-fold increment in the likelihood of having a diabetes diagnosis (*p* = 0.002−0.009), while subjects with low eGFR were 2.9-times more likely to be diagnosed with diabetes. These findings call for public measures to reduce environmental pollution by Cd and Pb and their food-chain transference to minimize Cd- and Pb-induced nephropathy.

## 5. Conclusions

We observed for the first time that people chronically exposed to Cd and Pb have enhanced risks of hyperglycemia, GFR reduction, and albuminuria. It is likely that one or both metals cause these adverse outcomes. The levels of exposure to environmental Cd and Pb among study subjects could be considered to be low to moderate, reflected by the arithmetic means (SD) for [Cd]_b_ and [Pb]_b_ of 0.59 (0.74) µg/L and 4.67 (4.88) µg/dL, respectively. However, the evidence linking these levels of environmental exposure to Cd and Pb to impaired glycemic control, reflected by abnormally high fasting plasma glucose levels, has emerged. Furthermore, plasma glucose levels above the kidney threshold for glucose of 180 mg/dL was associated with 5-fold increase in the POR for albuminuria. Research to elucidate the mechanism(s) underlying the association of abnormally high plasma glucose and exposure to Cd and Pb is warranted.

## Figures and Tables

**Figure 1 ijerph-19-02259-f001:**
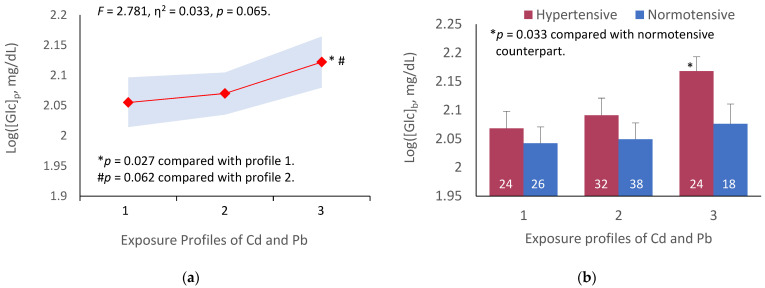
Association of cadmium and lead exposure profiles with fasting plasma glucose variation. The color-coded area graph depicts means for fasting plasma glucose levels as log[Glc]_p_ in participants with exposure profiles 1, 2, and 3 together with variability of the means (**a**). The bar graphs compare means for fasting plasma glucose as log[Glc]_p_ in participants with and without hypertension across three profiles (**b**). The number of participants with profiles 1, 2, and 3 are 50, 70, and 52, respectively. The means are obtained by univariate/covariance analysis, adjusted for age and BMI. Bonferroni correction is applied to multiple comparisons.

**Table 1 ijerph-19-02259-t001:** Characteristics of participants grouped by their cadmium and lead exposure profiles.

Variables	All Participants *n* = 176	Cadmium and Lead Exposure ^a^			*p*
		Profile 1, *n* 53	Profile 2, *n* 71	Profile 3, *n* 52	
Women, %	80.7	90.6	76.1	76.9	0.092
Diabetes diagnosis ^b^, %	50.0	35.8	46.5	69.2	0.002
Diabetes duration, years	9.3 ± 7.6	8.8 ± 7.8	9.2 ± 7.9	9.5 ± 7.4	0.910
Smoking, %	9.7	1.9	9.9	17.9	0.028
Hypertension ^c^ %	52.0	48.0	45.1	65.4	0.067
SBP, mmHg	138 ± 17	137 ± 17	136 ± 15	142 ± 19	0.122
DBP, mmHg	84 ± 9.5	83 ± 9.4	83 ± 8.7	86 ±10.4	0.226
Age, years	59.9 ± 9.7	59.5 ± 10.6	60.8 ± 9.2	58.9 ± 9.4	0.639
BMI, kg/m^2^	25.4 ± 4.7	25.8 ± 4.3	25.0 ± 4.0	25.6 ± 5.9	0.426
BMI classification ^d^					
Thin, %	6.8	5.7	4.2	11.5	0.472
Normal, %	28.4	26.4	32.4	25.0	0.162
Overweight, %	54.0	56.6	54.9	50.0	0.247
Obesity, %	10.8	11.3	8.5	13.5	0.949
[cr]_p_, mg/dL	0.87 ± 0.24	0.82 ± 0.15	0.89 ± 0.19	0.90 ± 0.36	0.133
[cr]_u_, mg/dL	89.2 ± 54.1	92.2 ± 51.7	90.1 ± 53.7	84.8 ± 57.9	0.499
[Alb]_u_, mg/dL	30.9 ± 79.9	23.8 ± 64.1	34.4 ± 90.2	33.8 ± 81.1	0.564
[Pb]_b_, µg/dL	4.67 ± 4.88	2.12 ± 0.00	4.58 ± 5.30	7.38 ±5.38	<0.001
[Cd]_b_, µg/L	0.59 ± 0.74	0.05 ± 0.05	0.65 ± 0.78	1.05 ± 0.72	<0.001
eGFR ^e^, mL/min/1.73 m^2^	79 ± 18	81 ± 16	77 ± 17	81 ± 21	0.269
≤60, %	16.5	11.3	22.5	13.5	0.196
ACR, mg/g creatinine	41.4 ± 103.8	28.4 ±75.8	38.5 ± 92.3	59.3 ± 139.2	0.387
<30, %	78.4	82.7	79.1	72.9	0.155
30−299, %	17.4	15.4	16.4	20.8	0.786
≥300, %	4.2	1.9	4.5	6.3	0.565
[Glc]_p_ ^f^ mg/dL	132 ± 61	118 ± 41	128 ± 66	150 ± 68	0.031
<110, %	49.4	58.5	52.1	36.5	0.055
110−126, %	11.9	15.1	12.7	7.7	0.368
127−179, %	25.6	17.0	26.8	32.7	0.155
≥180, %	13.1	9.4	8.5	23.1	0.154

SBP = systolic blood pressure; DBP = diastolic blood pressure; BMI = body mass index; cr = creatinine; Alb = albumin; [x]_p_ = plasma concentration of x; [x]_b_ = blood concentration of x; [x]_u_ = urine concentration of x; eGFR = estimated glomerular filtration rate; ACR = albumin-to-creatinine ratio; [Glc]_p_ = fasting plasma glucose concentration. ^a^ Profile 1 was defined as [Cd]_b_ *and* [Pb]_b_ ≤ its respective median of 0.3 µg/L and 2.12 µg/dL. Profile 2 was defined as [Cd]_b_ *or* [Pb]_b_ ≥ its respective median. Profile 3 was defined as [Cd]_b_ *and* [Pb]_b_ > its respective median. ^b^ Diabetes diagnosis was based on medical records. ^c^ Hypertension is defined as SBP ≥ 140 or DBP ≥ 90 mmHg. ^d^ BMI classification as thin, normal, overweight, and obesity correspond to BMI <18, 19–23, 24–30, and >30 kg/m^2^, respectively. ^e^ eGFR is determined with Chronic Kidney Disease Epidemiology Collaboration (CKD−EPI) equations. ^f^ [Glc]_p_ ≥ 110, 126, and 180 mg/dL indicate impaired fasting plasma glucose, diabetes diagnosis, and the renal threshold for glucose. Data for all continuous variables are arithmetic means ± standard deviation (SD). *p* ≤ 0.05 identifies statistical significance, determined with the Pearson Chi-Square test for % differences, and the Kruskal–Wallis test for mean differences.

**Table 2 ijerph-19-02259-t002:** Logistic regression analysis of type 2 diabetes diagnosis.

Independent Variables/Factors	Number of Subjects	Diabetes Diagnosis ^a^				
		β Coefficients (SE)	POR	95% CI		*p*
				Lower	Upper	
Age, years	173	0.014 (0.019)	1.015	0.977	1.054	0.457
BMI, kg/m^2^	173	−0.062 (0.040)	0.939	0.869	1.015	0.115
Smoking status	173	−0.951 (0.723)	0.386	0.094	1.593	0.188
Hypertension	173	−0.321 (0.334)	0.725	0.377	1.394	0.335
Gender	173	−0.450 (0.542)	0.638	0.220	1.846	0.407
eGFR, mL/min/1.73 m^2^						
>60	144	Referent				
≤60	29	1.074 (0.492)	2.926	1.115	7.679	0.029
Cd-Pb exposure ^b^						
Profile 1 ^c^	50	Referent				
Profile 2	71	1.096 (0.419)	2.991	1.317	6.794	0.009
Profile 3	52	1.431 (0.452)	4.182	1.725	10.14	0.002

POR = prevalence odds ratio; CI = confidence interval; BMI = body mass index; [Glc]_p_ = fasting plasma glucose level. Coding; non-smoker = 1; smokers = 2; hypertensive = 1; normotensive = 2; male = 1; female = 2. ^a^ A diagnosis of type 2 diabetes was based on a medical record of each subject. ^b^ Profile 1 was defined as [Cd]_b_ *and* [Pb]_b_ ≤ its respective median of 0.3 µg/L and 2.12 mg/dL. Profile 2 was defined as [Cd]_b_ *or* [Pb]_b_ ≥ its respective median. Profile 3 was defined as [Cd]_b_ *and* [Pb]_b_ > its respective median. ^c^ Blood pressure data were missing in three subjects.

**Table 3 ijerph-19-02259-t003:** Logistic regression modeling of abnormally high fasting plasma glucose levels.

Independent Variables/Factors	Model 1 ^a^, *n* 173		Model 2 ^b^, *n* 173		Model 3 ^c^, *n* 173	
	POR (95% CI)	*p*	POR (95% CI)	*p*	POR (95% CI)	*p*
Age, years	1.008 (0.975, 1.042)	0.653	1.033 (0.997, 1.069)	0.073	1.066 (1.011,1.124)	0.018
BMI, kg/m^2^	0.940 (0.875, 1.011)	0.095	0.958 (0.891, 1.029)	0.238	1.062 (0.959, 1.176)	0.250
Smoking status	0.567 (0.194, 1.662)	0.301	0.638 (0.209, 1.941)	0.428	0.246 (0.029, 2.063)	0.196
Cd-Pb exposure ^d^						
Profile 1	Referent		Referent		Referent	
Profile 2	1.940 (0.911, 4.132)	0.086	2.290 (1.073, 4.885)	0.032	3.383 (1.136, 10.08)	0.029
Profile 3	2.794 (1.228, 6.375)	0.014	2.964 (1.288, 6.822)	0.011	3.407 (1.061, 10.94)	0.039

[Glc]_p_ = fasting plasma glucose level; POR = prevalence odds ratio; BMI = body mass index. Coding; non-smoker = 1; smokers = 2. ^a^ Model 1 outcome = [Glc]_p_ ≥ 110 mg/dL. ^b^ Model 2 outcome = [Glc]_p_ ≥ 126 mg/dL. ^c^ Model 3 outcome = [Glc]_p_ ≥ 180 mg/dL. ^d^ Profile 1 was defined as [Cd]_b_ *and* [Pb]_b_ ≤ its respective median. Profile 2 was defined as [Cd]_b_ *or* [Pb]_b_ ≥ its respective median. Profile 3 was defined as [Cd]_b_ *and* [Pb]_b_ > its respective median.

**Table 4 ijerph-19-02259-t004:** Bivariate correlation analysis of fasting plasma glucose levels.

Variables	Pearson’s Correlation Coefficients								
	[Glc]_b_	Age	BMI	Gender	SMK	[Pb]_b_	[Cd]_b_	ACR	eGFR
[Glc]_b_	1								
Age	−0.193 *	1							
BMI	0.078	−0.285 ***	1						
Gender	−0.057	0.186 *	0.140 #	1					
SMK	−0.040	−0.018	−0.152 *	−0.620 ***	1				
[Pb]_b_	0.194 *	−0.071	0.060	−0.108	0.153 *	1			
[Cd]_b_	0.049	0.053	−0.047	−0.034	0.184 *	0.148 *	1		
ACR	0.288 ***	0.114	0.003	0.009	−0.031	0.047	0.098	1	
eGFR	0.092	−0.454 ***	0.172 *	−0.139	0.040	0.090	−0.123	−0.176 *	1
Cd/Pb profiles	0.216 **	−0.022	−0.017	−0.134	0.202 **	0.604 **	0.724 ***	0.102	−0.008

[Glc]_b_ fasting blood glucose levels as log [Glc]_b_; BMI = body mass index; SMK = smoking status; [Pb]_b_ = blood concentration of Pb as log([Pb]_b_) × 100; [Cd]_b_ = blood concentration of Cd as log([Cd]_b_) × 100; ACR as log[(ACR) × 10^3^]. Coding: male = 1; female = 2; nonsmoker = 1; smoker = 2; profile 1 = 1; profile 2 = 2; profile 3 = 3. *p*-values ≤ 0.05 indicate statistically significant correlation of individual pairs of variables. # *p* = 0.054; * *p* = 0.010−0.050; ** *p* = 0.004−0.007; *** *p* < 0.001.

**Table 5 ijerph-19-02259-t005:** Association of the prevalence odds for albuminuria with abnormal fasting plasma glucose levels.

Independent Variables/Factors	Number of Subjects ^b^	Albuminuria ^a^				
		β Coefficients (SE)	POR	95% CI		*p*
				Lower	Upper	
Age, years	164	−0.040 (0.028)	0.961	0.910	1.014	0.147
BMI, kg/m^2^	164	−0.027 (0.047)	0.974	0.888	1.068	0.570
Smoking status	164	−1.287 (0.914)	0.276	0.046	1.656	0.159
Hypertension	164	−1.225 (0.482)	0.294	0.114	0.755	0.011
Gender	164	−1.825 (0.644)	0.161	0.046	0.570	0.005
eGFR, mL/min/1.73 m^2^						
>60	138	Referent				
≤60	26	1.129 (0.575)	3.093	1.002	9.552	0.050
[Glc]_p_, mg/dL						
<110	81	Referent				
110−126	20	0.453 (0.634)	1.574	0.454	5.457	0.475
127−179	43	1.210 (0.816)	3.353	0.678	16.58	0.138
≥180	20	1.597 (0.653)	4.937	1.373	17.76	0.014

^a^ Albuminuria is defined as albumin to creatinine ratio ≥ 20 mg/g for men, and ≥30 mg/g for women. ^b^ Urine albumin or blood pressure data were missing in 12 participants. POR = prevalence odds ratio; CI = confidence interval; BMI = body mass index; [Glc]_p_ = fasting plasma glucose level. Coding; non-smoker = 1; smokers = 2; hypertensive = 1; normotensive = 2; male = 1; female = 2.

## Data Availability

All data are contained within this article.

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
