# Peer review of "Effects of Environmental Exposure to Cadmium and Lead on the Risks of Diabetes and Kidney Dysfunction"

_ijerph, 2022, doi:10.3390/ijerph19042259_

Round 1
Reviewer 1 Report
Major points
- The authors are suggested to describe the importance in diabetes that occurs related to exposure, but also the association with average age.
- In methodology, the authors should mention the reason of differ Cadmium and Lead Exposure Profiles (low, moderate, and high), not in results and their importance. This profiles equally impacts kidney function and diabetes risk
- Because endocrine disruptors (Cd and Pb) that are being evaluated can trigger several tissue damages, the authors should include the association with the immune system (for example a marker in serum or a marker of tubular damage in urine associated with inflammation and toxins, markers more sensitive than creatinine or albumin)
Minor points
- In abstract, the authors should mention that is abbreviature POR
Author Response
Reviewer 1
We thank the Reviewer for the comments and suggestions. We have revised our manuscript accordingly. We have provided below point-by-point responses to issues/concerns raised.
Major points
Point 1: The authors are suggested to describe the importance in diabetes that occurs related to exposure, but also the association with average age.
Response: We have inserted in Table 1 the duration of diabetes among participants. In addition, we have exemplified the importance of age in the present study (lines 238-241), as quoted below.
“The means for fasting plasma glucose were adjusted for age and BMI. Because Cd and Pb both are cumulative toxicants, the body burden of these metals increases with age. Obesity (BMI>30 kg/m2) is a known risk factor of diabetes and hypertension.”
Point 2: In methodology, the authors should mention the reason of differ Cadmium and Lead Exposure Profiles (low, moderate, and high), not in results and their importance. This profiles equally impacts kidney function and diabetes risk.
Response: We have provided the reason for grouping of subjects in the Subsection 2.4 (lines 111-120), quoted below. We have indicated in abstract, the equal effect of Cd-Pb exposure on kidney function and diabetes risk.
2.4. Toxic Metal Exposure Profiling
To enable evaluation of the impacts of simultaneous exposure to Cd and Pb on the risks of diabetes and kidney dysfunction, exposure to these toxic metals was based on [Cd]b and [Pb]b. Each subject was assigned to the exposure profile 1, 2 or 3 by comparing her/his [Cd]b and [Pb]b with the median for [Cd]b of 0.3 µg/L and the median for [Pb]b of 2.12 µg/dL. Given the sample size of 176, we used the median as a cutoff value to obtain subgroups with sufficient numbers of participants. Exposure profile 1 was defined as [Cd]b and [Pb]b levels ≤ medians. Exposure profile 2 was defined as [Cd]b or [Pb]b levels ≥ medians. Exposure profile 3 was defined as [Cd]b and [Pb]b levels > medians. There were 53, 71 and 52 subjects with exposure profiles 1, 2 and 3, respectively.
Point 3: Because endocrine disruptors (Cd and Pb) that are being evaluated can trigger several tissue damages, the authors should include the association with the immune system (for example a marker in serum or a marker of tubular damage in urine associated with inflammation and toxins, markers more sensitive than creatinine or albumin).
Response: We thank the Reviewer for the suggestion to examine markers in plasma and/or urine that better probe effects of Cd and Pb as endocrine disrupting chemicals. We will take on the suggestion in our future works.
Minor points
In abstract, the authors should mention that is abbreviature POR.
Response: The word “POR” appearing in abstract is now fully spelt.

Reviewer 2 Report
Abstract
Line 22- Where is written “these nephrotoxic metals on the risks of diabetes, and kidney malfunction, assessed by albuminuria, and the estimated glomerular filtration rate”, the order of the contents if confusing.
Line 30 – Spell out “POR”.
Introduction
Line 43 – I suggest substitute the word “characterized” for another one like detected/assessed.
Line 61 – Spell out “ACR”.
Methods
Line 75 – I believe that the information “and those who denied an invitation to participate” is not necessary.
Line 81 – Please provide a brief information explained why and how the hypertension factor was included in this study; the same in line 225.
Line 109 – Please provide a reference for this procedure: “When [Cd]b and [Pb]b levels were less than their detection limits, the 109 concentrations assigned were their detection limits divided by the square root of 2”.
Line 132 – The sentence “[Glc]p levels ≥ 126 mg/dL, [Glc]p levels ≥ 180 mg/dL 131 and albuminuria, defined as a ACR ≥ 20 mg/g for men and ACR ≥ 30 mg/g for women” is not clear.
Line 134 – Please explain the reason for adjusting for age and BMI. Why were these variables chosen and for e.g. “smoking” was not?
Table 1 – Agreeing the criteria used to define the low, moderate and high exposure groups, I suggest in further works the use of cluster analysis since will certainly improve the analysis of the data. Considering other for studies on thresholds for toxic effects and general exposures, I also wonder if the terms “low”, “moderate” or “high” are correct. Paradoxically, in line 330 the authors consider that all the exposures were “low environmental exposures”. Perhaps other terminology such as level 1, 2 or 3 would be more appropriate.
Line 158 – Please refer the BMI class in “mean body mass index (BMI) was 25.4 kg/m2”. The indication of underweight etc. makes the interpretation easier for the reader.
Line 251 – Where is written “joint effects of simultaneous exposure to Cd and Pb”, it would be more cautious to mention “effects of simultaneous exposure to Cd and Pb” since there is what the data provides. In this study synergism/additivity is not assessed; the same for line 22 for the word “combined”.
Line 284 - Some caution should be undertaken in the interpretation of the results. It is mentioned “Nevertheless, a lack of a positive association between Cd exposure estimates and body weight seen in the present study is consistent with results of studies in the U.S”. A correlation between Cd and effects is not enough. How can the authors differentiate if the effects are attributable to Cd or to the co-exposure to Cd and Pb with the available data? Was Pb fixed as a confounder in the correlation analysis for Cd? Similar rationales are present along the paper. However, it does not mean that a rational like “consequence of Cd toxicity in proximal tubular cells, known to accumulate Cd” in line 323 or in line 335 “Data from a Korean population study suggested that con-335 current Pb exposure may enhance the nephrotoxicity induced by Cd” is incorrect.
Author Response
Reviewer 2
We thank the Reviewer for the comments and suggestions. We have revised our manuscript accordingly. We have provided below point-by-point responses to issues/concerns raised.
Abstract
Line 22- Where is written “these nephrotoxic metals on the risks of diabetes, and kidney malfunction, assessed by albuminuria, and the estimated glomerular filtration rate”, the order of the contents if confusing.
Response: We have reworded the statement in the text. It now reads as below.
“The aim of this study was to examine the effects of concurrent exposure to these toxic metals on the risks of diabetes and kidney functional impairment.”
Line 30 – Spell out “POR”.
Response: We have spelt out the word “POR” in abstract.
Introduction
Line 43 – I suggest substitute the word “characterized” for another one like detected/assessed.
Response: We have changed the sentence to read as quoted below.
“The diagnosis of CKD is based on albumin-to-creatinine ratio (ACR) above 30 mg/g creatinine (albuminuria) and/or a decrease of the glomerular filtration rate (GFR) to 60 mL/min/1.73 m2 (low eGFR) that persists for at least three months [16-19].”
Line 61 – Spell out “ACR”.
Response: We have spelt out “ACR” appearing on line 61.
Methods
Line 75 – I believe that the information “and those who denied an invitation to participate” is not necessary.
Response: The phrase has been deleted.
Line 81 – Please provide a brief information explained why and how the hypertension factor was included in this study; the same in line 225.
Response: We have explained the reason for inclusion hypertension in the covariance analysis of fasting plasma glucose variation (Figure 1) as quoted below (lines 238-241).
“The means for fasting plasma glucose were adjusted for age and BMI. Because Cd and Pb both are cumulative toxicants, the body burden of these metals increases with age. Obesity (BMI>30 kg/m2) is a known risk factor of diabetes and hypertension.”
Line 109 – Please provide a reference for this procedure: “When [Cd]b and [Pb]b levels were less than their detection limits, the concentrations assigned were their detection limits divided by the square root of 2”.
Response: A reference for the assignment of values to blood samples containing nondetectable levels of Cd and Pb has been inserted as quoted below.
[38[ Hornung, R.W.; Reed, L.D. Estimation of average concentration in the presence of nondetectable values. Appl. Occup. Environ. Hyg. 1990, 5, 46-51.
Line 132 – The sentence “[Glc]p levels ≥ 126 mg/dL, [Glc]p levels ≥ 180 mg/dL 131 and albuminuria, defined as a ACR ≥ 20 mg/g for men and ACR ≥ 30 mg/g for women” is not clear.
Response: We have rewritten the description for dichotomized variables (lines 139-143) as quoted below.
“Abnormal fasting plasma glucose was defined as [Glc]p levels ≥ 110 mg/dL. Diabetes was diagnosed when fasting [Glc]p levels was ≥ 126 mg/dL. The renal threshold for glucose was assumed to be [ Glc]p ≥ 180 mg/dL. Albuminuria was defined as a ACR ≥ 20 mg/g for men and ≥ 30 mg/g for women.”
Line 134 – Please explain the reason for adjusting for age and BMI. Why were these variables chosen and for e.g. “smoking” was not?
Response: We have provided the reasons for adjustment of the means for fasting plasma glucose for age and BMI, but not for smoking (lines 236-241), as quoted below.
“Because the prevalence of smoking among participants was low (9.7%) and there was a gender bias (of 18 smokers, one was woman), the mean for fasting plasma glucose derived for each Cd-Pb exposure profile was not adjusted for smoking. The means for fasting plasma glucose were adjusted for age and BMI. Because Cd and Pb both are cumulative toxicants, the body burden of these metals increases with age. Obesity (BMI>30 kg/m2) is a known risk factor of diabetes and hypertension.”
Results and Discussion
Table 1 – Agreeing the criteria used to define the low, moderate and high exposure groups, I suggest in further works the use of cluster analysis since will certainly improve the analysis of the data. Considering other for studies on thresholds for toxic effects and general exposures, I also wonder if the terms “low”, “moderate” or “high” are correct. Paradoxically, in line 330 the authors consider that all the exposures were “low environmental exposures”. Perhaps other terminology such as level 1, 2 or 3 would be more appropriate.
Response: We thank the reviewer for the suggestion regarding data analysis. We will take on the cluster analysis in our future works. Our description of exposure to Cd and Pb as low, moderate or high have been changed to profile 1, profile 2 and profile 3 (Table 1). We have made changes throughout the text where applicable.
Line 158 – Please refer the BMI class in “mean body mass index (BMI) was 25.4 kg/m2”. The indication of underweight etc. makes the interpretation easier for the reader.
Response: We have inserted BMI classification in Table 1 together with the % subjects across three Cd-Pb exposure profiles. We have explained these BMI data in the text (lines 164-165, lines 177-178)
Line 251 – Where is written “joint effects of simultaneous exposure to Cd and Pb”, it would be more cautious to mention “effects of simultaneous exposure to Cd and Pb” since there is what the data provides. In this study synergism/additivity is not assessed; the same for line 22 for the word “combined”.
Response: Where applicable, we have removed “joint”, “combined” to describe our work. We use “simultaneous” or “concurrent” exposure instead.
Line 284 - Some caution should be undertaken in the interpretation of the results. It is mentioned “Nevertheless, a lack of a positive association between Cd exposure estimates and body weight seen in the present study is consistent with results of studies in the U.S”. A correlation between Cd and effects is not enough. How can the authors differentiate if the effects are attributable to Cd or to the co-exposure to Cd and Pb with the available data? Was Pb fixed as a confounder in the correlation analysis for Cd? Similar rationales are present along the paper. However, it does not mean that a rational like “consequence of Cd toxicity in proximal tubular cells, known to accumulate Cd” in line 323 or in line 335 “Data from a Korean population study suggested that concurrent Pb exposure may enhance the nephrotoxicity induced by Cd” is incorrect.
Response 1: We thank the author for raising this issue. For clarity, we have rewritten this part of discussion (lines 282-300). Please kindly note that the discussions in this paragraph are pertaining to the logistic regression analysis shown in Tables 2 and 3. It is important to note that an inverse association between Cd exposure measures and BMI plus other obesity measures have consistently been seen across populations of adults and children. In revision, references for such inverse relationship in children have been added (lines 297-300, references 44, 45). We have explained the implication of the observation on an inverse association of Cd and obesity (lines 317-321), quoted below.
“It is noteworthy that the worldwide rising incidence of type 2 diabetes mellitus has often been linked to increasing prevalence of obesity, but studies in various populations, described above, found an inverse association between Cd exposure estimates and body weight gain and other measures of obesity. Consequently, exposure to Cd, especially of dietary origin, appears to be a contributor to the global increase in prevalence of diabetes.”
[44] Dhooge, W.; Hond, E.D.; Koppen, G.; Bruckers, L.; Nelen, V.; Van De Mieroop, E.; Bilau, M.; Croes, K.; Baeyens, W.; Schoeters, G.; Van Larebeke, N. Internal exposure to pollutants and body size in Flemish adolescents and adults: associations and dose-response relationships. Environ. Int. 2010. 36, 330-337.
[45] Shao, W.; Liu, Q.; He, X.; Liu, H.; Gu, A.; Jiang, Z. Association between level of urinary trace heavy metals and obesity among children aged 6-19 years: NHANES 1999-2011. Environ. Sci. Pollut. Res. Int. 2017, 24, 11573-11581.
Response 2. The pathogenesis of Cd-induced GFR reduction described in the text are based on published research data (references 49 and 50).
Response 3: The referred statement (line 335) was deleted as was the related reference # 48.
